# RF-MEMS Monolithic K and Ka Band Multi-State Phase Shifters as Building Blocks for 5G and Internet of Things (IoT) Applications

**DOI:** 10.3390/s20092612

**Published:** 2020-05-03

**Authors:** Jacopo Iannacci, Giuseppe Resta, Alvise Bagolini, Flavio Giacomozzi, Elena Bochkova, Evgeny Savin, Roman Kirtaev, Alexey Tsarkov, Massimo Donelli

**Affiliations:** 1Center for Materials and Microsystems (CMM), Fondazione Bruno Kessler (FBK), Via Sommarive, 18-38123 Trento, Italy; resta@fbk.eu (G.R.); bagolini@fbk.eu (A.B.); giaco@fbk.eu (F.G.); 2Bazovye Tekhnologii, LLC, 125124 Moscow, Russia; elena.bochkova@btlabs.ru (E.B.); e.savin@btlabs.ru (E.S.); roman.kirtaev@phystech.edu (R.K.); a.tsarkov@btlabs.ru (A.T.); 3Department of Information Engineering and Computer Science, University of Trento, 38123 Trento, Italy; massimo.donelli@unitn.it

**Keywords:** RF-MEMS, MEMS, RF passives, phase shifters, multi-state passive networks, 5G, Internet of Things (IoT), wideband operability

## Abstract

RF-MEMS, i.e., Micro-Electro-Mechanical Systems (MEMS) for Radio Frequency (RF) passive components, exhibit interesting characteristics for the upcoming 5G and Internet of Things (IoT) scenarios, in which reconfigurable broadband and frequency-agile devices, like high-order switching units, tunable filters, multi-state attenuators, and phase shifters will be necessary to enable mm-Wave services, small cells, and advanced beamforming. In particular, satellite communication systems providing high-speed Internet connectivity utilize the K and Ka bands, which offer larger bandwidth compared to lower frequencies. This paper focuses on two design concepts of multi-state phase shifter designed and manufactured in RF-MEMS technology. The networks feature 4 switchable stages (16 states) and are developed for the K and Ka bands. The proposed phase shifters are realized in a surface micromachining RF-MEMS technology and the experimentally measured parameters are compared with Finite Element Method (FEM) multi-physical electromechanical and RF simulations. The simulated phase shifts at both the operating bands fit well the measured value, despite the measured losses (S21) are larger than 5–7 dB if compared to simulations. However, such a non-ideality has a technological motivation that is explained in the paper and that will be fixed in the manufacturing of future devices.

## 1. Introduction

RF-MEMS, i.e., Micro-Electro-Mechanical Systems (MEMS) for Radio Frequency (RF) applications have been around for a long time. First discussed in scientific papers after the mid-1990s [1,2], MEMS-based RF passives quickly ignited huge expectations in terms of market breakthroughs, primarily because of their remarkable and unprecedented electromagnetic characteristics, like wideband operability, large reconfigurability/tunability, very-low loss, high-isolation, high Q-factor [3].

Despite the initial market expectations of RF-MEMS were disappointed, the context started to change quite radically after 2010 when smartphones started to become increasingly more popular. The increasing integration of devices in the handsets led to a progressive worsening of the quality of communications [4,5]. Analog impedance matching tuners based on RF-MEMS technology were the first in line solution available [6], which made possible solving the problem, and, on the other hand, to score the first successful exploitation of RF-MEMS in mass-market applications. Moreover, the spread of RF-MEMS-based reconfigurable passive networks in the consumer segment is pulling applied research and engineering of other components based on Microsystem technology, with robust switches and switching units first in line [7,8].

Besides the abovementioned case, another intriguing application field for high-performance RF-MEMS is gaining shape, which is the upcoming fifth generation of mobile communications, i.e., 5G, with all its implications and spillovers in the Internet of Things (IoT) scenario.

Active development of 5G technologies and satellite communications demand for high-speed stable data transfer services from any location. Satellite systems providing high-speed Internet connectivity utilize K and Ka frequency bands offering a large bandwidth compared to lower frequencies. Conventionally, signal reception and transmission are carried out through two separate frequency channels in the K and Ka frequency bands, which, in turn, increase the demand for the high-performance components intended for signal control of the transceiver modules.

In light of this scenario, this work presents and discusses a couple of design concepts of RF-MEMS-based reconfigurable phase shifters. From the architectural point of view, we relied on the True-Time Delay (TTD) topology, which is well-known in the literature. Given the field of application mentioned above, we tailored the design of the network in such a way to have the center frequency of operation at 19.5 GHz and 29.3 GHz, i.e., in the K and Ka band, respectively. Concerning reconfigurability, we opted for 4 stages (i.e., 16 configurations), controlled by electrostatically driven ohmic series RF-MEMS micro-relays. In particular, the K band phase shifter employs a classical transversal clamped-clamped switch topology. On the other hand, the Ka band phase shifters, given its higher operating frequency, imposes more stringent requirements in terms of reduced network footprint. Having in mind such a driver, we developed and implemented an innovative in-line longitudinal RF-MEMS switch design, conjugating the compactness typical of cantilevered (single-hinged) micro-relays, with the mechanical robustness and resilience of clamped-clamped solutions.

The paper is arranged as follows. After the current section framing the evolution of RF-MEMS and the opportunities of the 5G and IoT application fields, the following Section 2 will introduce the reconfigurable phase shifters design concepts, along with modeling and simulations. Section 3 will develop a comprehensive discussion of the RF-MEMS phase shifters, leveraging the acquired experimental data. Finally, Section 4 will collect some conclusive considerations.

## 2. RF-MEMS Phase Shifter: Design Concept and Simulations

The RF-MEMS phase shifters concepts operating at K and Ka frequency bands are designed capitalizing on True-Time Delay (TTD) technology [9,10,11,12,13,14,15] and are manufactured in the surface micromachining technology available at Fondazione Bruno Kessler (FBK) in Italy [16]. The resulting phase delay is defined by the lengths of the transmission lines. By selecting different propagating signal paths in the circuit, the total phase shift can be tuned by a predefined value. The designed phase shifters consist of 4 sections (i.e., 4 bits), connected in series. Each section has two RF paths with different lengths, as reported by the schematic in Figure 1.

The total phase shift is controlled by the combination of the four sections path lengths, selected using RF-MEMS ohmic switches monolithically integrated into the network. Once the switches are actuated (i.e., pulled-in), the RF signal travels through the selected delay line. Each bit can provide an additional 22.5 degrees phase shift at the operating frequency. Thus, a 4 bits phase shifter provides 2^4^ = 16 different phase delays: 0°, ±22.5°, ±45°, ±67.5°, ±90°, ±112.5°, ±135°, ±157.5°, and 180° at 19.5 GHz and 29.3 GHz center frequencies, for the K and Ka band specific design variant, respectively.

The designed phase shifter is based on 50 Ω Coplanar Waveguide (CPW) transmission line and the time delay switch option is based on series-type electrostatically driven RF-MEMS ohmic switches. Thus, the phase shift is controlled by the applied voltage signal to each of the four bits. The K band phase shifter is fitted with switches described by the authors in [17]. Figure 2a shows a schematic view of the transverse RF-MEMS switching unit integrated into the CPW transmission line.

The Ka band device is almost twice smaller in size, so the same switch geometry cannot be fitted in it due to its dimensions and space needed for integration. One of the most compact configurations is related to cantilever-type geometry (i.e., single-hinged membrane) [18]. However, from a general point of view, cantilevered MEMS switches are not as robust as clamped-clamped designs [19,20]. To avoid reliability issues, a compact longitudinal membrane with complex anchor geometry has been designed, as reported in Figure 2b. This membrane with optimized characteristics allows lowering the actuation voltage and reducing the area occupation, while still relying on double-hinged MEMS suspended membrane solution, as for the transverse micro-switch.

Both the longitudinal and transverse switch geometries were optimized to achieve a wide operating frequency range in terms of low loss and high isolation. Table 1 shows that in the operating range 0–40 GHz, the simulated losses are less than −0.8 dB and isolation is better than −20 dB.

Structural deformations due to the actuation voltage were simulated for 1.5 µm membrane thickness and stiffening rib [16] of 2.5 µm at varied air-gaps between the membrane and the actuation pads. Table 2 reports the applied actuation voltage depending on the air-gap for both membranes.

The results of electromechanical simulations for the minimum gap of 2 µm are illustrated in Figure 3, in which the actuation voltage (*V_PI_*) is 28 V for the transverse MEMS membrane and 38 V for the novel compact longitudinal design concept.

The RF performance of the phase shifters employing the just discussed micro-relays is simulated by means of Finite Element Method (FEM) analysis. Figure 4 and Figure 5 show the reflection (S11), insertion loss (S21), and phase delay for several configurations of the K and Ka band phase shifters, respectively. The simulated losses at 19.5 GHz are better than −2 dB in a 2 GHz frequency span, while reflection is between −40 and −25 dB at the center frequency (K band phase shifter; see Figure 4).

On the other hand, the device working at 29.3 GHz has simulated losses better than −2.4 dB for most configurations, despite at −90 and +90 phase shifts, −2.6 dB losses are observed at the maximum limit of 2 GHz frequency range (Ka band phase shifter; see Figure 5).

As conclusive consideration of this section, it must be highlighted that most of the losses are due to microswitches, as it will be discussed more in technical details later in this paper, especially for what concerns the non-idealities exhibited by the tested physical samples.

## 3. Experimental Results and Discussion

In this section, an extensive experimental characterization is reported, both for what concerns the RF-MEMS phase shifters physical samples, as well as for the standalone micro-switches they employ. The mentioned testing comprises the analysis of the RF characteristics and the electromechanical behavior of the available RF-MEMS samples.

The S-parameters (scattering parameters) are measured using a probe station equipped with Ground-Signal-Ground (GSG) CPW microprobes (250 μm pitch), a Vector Network Analyzer (VNA) with an operating frequency range from 10 kHz up to 40 GHz and a voltage source. The VNA is calibrated with the SOLT (Short Open Load Thru) method using an on-wafer GSG calibration kit. The discussed phase shifters require actuating four DC signal pads and one ground (GND) pad. Thus, we exploit a setup featuring five DC probes and two RF probes to fully characterize the physical devices. The probes setup is shown in Figure 6.

In the first place, the attention is concentrated on the standalone RF-MEMS switching unit electromagnetic characteristics. In this regard, the transverse and longitudinal ohmic micro-relays previously discussed in Section 2 and reported in Figure 2 and Figure 3 are measured up to 40 GHz. The following Figure 7 and Figure 8 show the comparison of measured and simulated S-parameters of transverse and longitudinal switches, respectively, in the OFF and ON state.

Concerning the isolation (S21) in the OFF state, the simulated traces predict quite accurately the experimental behavior, as visible in Figure 7b and Figure 8b. A more articulated discussion has to be developed for the ON state of both switches as the disagreement between measurements and simulations is relevant. In particular, the measured reflection (S11) is worse than simulations for both switches, as emerged in Figure 7c and Figure 8c, with particular reference to the latter case (longitudinal switch). Relevantly, the losses (S21) of the physical samples are significantly larger than expected, being them worse than simulations of an extent ranging from 0.2–0.3 dB up to 0.8–0.9 dB. In particular, the measured S21 of the transverse switch employed in the K band phase shifter is −0.8 dB at 19.5 GHz, while the simulated prediction is −0.1 dB (see Figure 7d). The experimental S21 of the longitudinal switch employed in the Ka band phase shifter is −1 dB at 29.3 GHz while the simulation predicts a value slightly larger than −0.1 dB (see Figure 8d).

Despite such a rather severe performance drift in terms of larger attenuation, its explanation is straightforward and the solution quite easy as the cause is related to a known technology aspect. The physical samples available at the time of this paper’s writing are coming from the first wafer of the RF-MEMS fabrication batch processed in the facility. In such a wafer, which was considered a preliminary test, strict control on the cleaning of through-oxide vias to the aluminum underpass (after their opening) was not performed [16]. This happened for a technical problem and led to larger contact resistance between gold and aluminum, detected when measuring a few test structures for process control. This non-ideality also affects S-parameters leading to the bad performance shown in Figure 7c,d and Figure 8c,d for the switches in the ON state.

In the next wafers of the same fabrication batch, still to be finalized, the issue of vias opening was correctly performed and we expect significantly lower values of contact resistance and, in turn, reduced losses. To provide an experimental indication of the low-losses typically achieved by the RF-MEMS technology employed for implementing the phase shifters here reported, it can be helpful referring to [8,19]. In such works, RF-MEMS series ohmic switches with through-oxide vias properly opened and cleaned showed measured values of the S21 parameter better than −1 dB up to 50 GHz when conducting (ON state) and, more importantly, better than −0.3 dB up to 30 GHz.

Of course, the employment of switching units with the increased losses just reported affects negatively the RF characteristics of the phase shifters. In this respect, one should bear in mind that, regardless of the particular configuration of 4 bits, the RF signal has to travel through eight ON switches. Therefore, the S11 and S21 simulated characteristics previously shown in Figure 4a,b and Figure 5a,b cannot be met by the physical samples, as it is going to be discussed in the following.

To conclude the discussion on the testing of standalone switches, it must be reported that the DC characterization of the pull-in behavior of the physical samples showed a satisfactory agreement with the actuation voltages previously discussed in Table 1 and Table 2. This consideration applies to all the tested replicas of the transversal and longitudinal RF-MEMS micro-relays, which showed variations of the pull-in voltage always in the range of a few percent with respect to simulations. Further details are not provided for the sake of brevity, leaving proper room to the upcoming RF characterization of the phase shifters.

The 4-bit phase shifters, regardless of the operating frequency band, have four actuation pads for each reference line, four pads for the respective four different phase delays and a single ground pad (labeled as GND). A layout top-view of both devices is provided in Figure 9.

The reference phase pads are labeled as S_0_ and placed at the top of the scheme. The pads named S_22.5_, S_45_, S_90,_ and S_180_ route the DC signal to the branches realizing the corresponding phase delays. The K and Ka bands RF-MEMS phase shifters have a silicon chip footprint of 10.5 mm × 7.5 mm and 7.5 mm × 8.5 mm, respectively.

The desired bits combinations are obtained imposing a 45 V DC signal to the corresponding pads (i.e., above pull-in) and 0 V to the ground pad. For the sake of clarity, in the following, we show measurements for just a few bits combinations, namely 0, 45, 67.5, 90, and 180 degrees of phase shift. We measured the reflection (S11), insertion loss (S21), and phase delay in a 2 GHz frequency range for the K and Ka band operating frequencies, taking into account the diversity of the two proposed design concepts. The S-parameters measurements are reported in the following Figure 10 and Figure 11, for the K and Ka band phase shifters, respectively.

The measured reflection of the observed configurations ranges between −15 and −25 dB at the center frequency (see Figure 10a), while the simulated values are comprised between −25 dB and −43 dB (see Figure 4a). More relevantly, the measured transmission is between −6.8 and −8.6 dB at the center frequency (see Figure 10b), while the simulations show an S21 between −1.4 and −1.8 dB (see Figure 4b). On the other hand, the measured phase delay is more accurate, as it will be discussed shortly later.

For the Ka band network, the measured S11 ranges between −13 and −17 dB at the center frequency (see Figure 11a), while the simulated values are comprised between −14 dB and −35 dB (see Figure 5a). The measured transmission is between −7 and −9.5 dB at the center frequency (see Figure 11b), while the simulations show an S21 between −1.7 and −2.4 dB (see Figure 5b). On the other hand, as for the previous network, the measured phase delay is more accurate.

For what concerns the characteristics of reflection and transmission of both the phase shifters designs, the influence of the switches non-idealities previously reported leads to a significant worsening of the performance of the network, which is particularly severe for the losses. Nonetheless, the fact that the worsening of experimental S-parameters is rather independent on the particular network configuration and switch design corroborates the explanation that it is due to the additional contact resistance of the switches. Eventually, as discussed before, the technological non-ideality was already solved in the next wafers still to be finalized.

In conclusion, despite the large divergence between experimental and nominal characteristics, the measured phase delay configuration exhibits a satisfactory match with the simulations, as highlighted by the phase difference plots reported in Figure 12, for a very narrow range around the K and Ka band center frequencies (Figure 12a,b, respectively).

## 4. Conclusions

RF-MEMS, i.e., Microsystems (MEMS) based Radio Frequency passive components, exhibit pronounced characteristics in terms of high performance, tunability/reconfigurability, and frequency agility, making them an interesting solution for the upcoming fifth generation of mobile communications, i.e., 5G, with all its implications and spillovers in the Internet of Things (IoT). Active development of 5G technologies and satellite communication demands for high-speed stable data transfer services from any location. Satellite systems providing high-speed Internet connectivity utilize K and Ka frequency bands offering a larger bandwidth compared to lower frequencies bands.

This work presented and discussed a couple of design concepts of RF-MEMS-based reconfigurable phase shifters working in the K and Ka bands, with 19.5 GHz and 29.3 GHz center frequency, respectively. Both networks feature 4 reconfigurable stages and realize 16 different phase delays by means on series ohmic RF-MEMS switches. In particular, the phase shifter design concept working in the Ka band employs a novel in-line switch solution, which allows conjugating the compactness of cantilevered micro-relays with the robustness of the clamped-clamped configuration.

The characteristics and performance of the RF-MEMS phase shifters and standalone switches were extensively discussed, capitalizing on Finite Element Method (FEM) simulations and experimental testing of the S-parameters.

Due to a technology-related non-ideality, the ON-state resistance of the switches was larger than the nominal value. This led to losses around 0.8–0.9 dB at K and Ka center frequencies, while the simulations predicted 0.1–0.2 dB. As the discussed phase shifters, regardless of the selected configuration, always employ eight cascaded ON switches, the impact on the overall RF characteristics is important. The measured networks exhibited additional losses (S21) as large as 5–7 dB at the two center frequencies when compared to FEM simulations. However, the reason for such a scarce performance is known and identified at the technology level, and the next fabricated samples will show better performance. Finally, despite the large divergence from the nominal specifications, the measured phase shift configurations exhibited limited differences when compared to simulations.

## Figures and Tables

**Figure 1 sensors-20-02612-f001:**
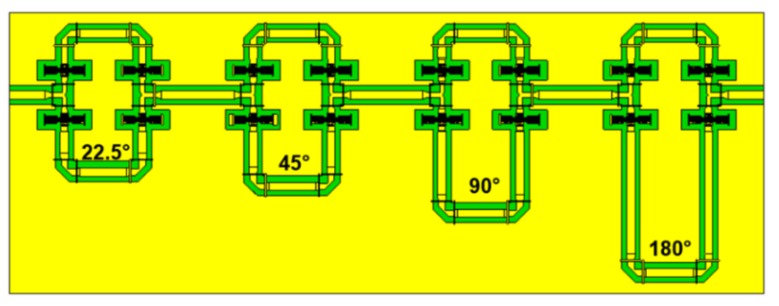
Top view of the 4-bit True-Time Delay (TTD) phase shifter schematic topology. The upper sections correspond to the reference path with a total 0° phase shift.

**Figure 2 sensors-20-02612-f002:**
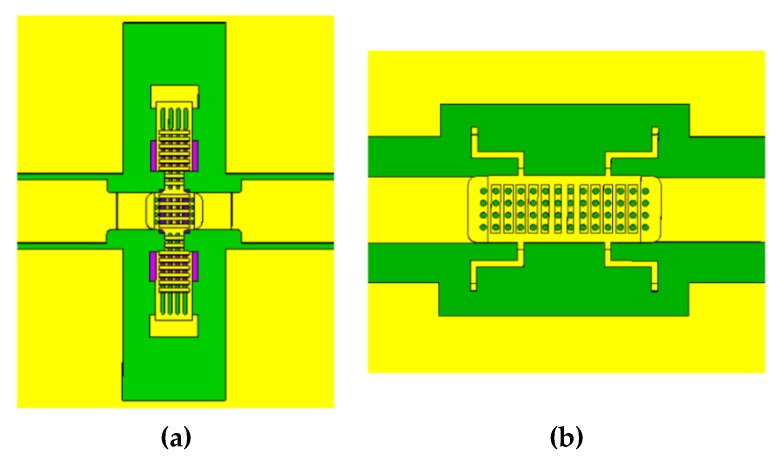
Schematic of the transverse (**a**) and longitudinal (**b**) Micro-Electro-Mechanical Systems (MEMS) membrane integrated into the Coplanar Waveguide (CPW) transmission line and employed in the K and Ka band phase shifters designs, respectively.

**Figure 3 sensors-20-02612-f003:**
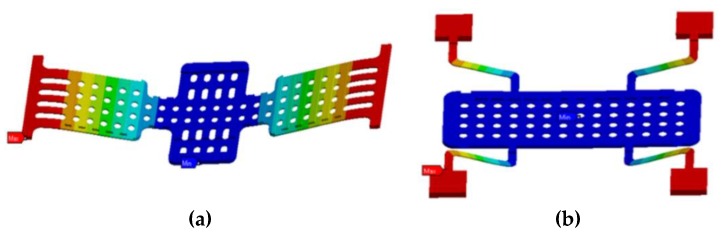
Electromechanical simulated results for the transverse (**a**) and longitudinal (**b**) RF-MEMS membrane geometries with 2 µm air-gap. The color scale indicates the membrane deformation.

**Figure 4 sensors-20-02612-f004:**
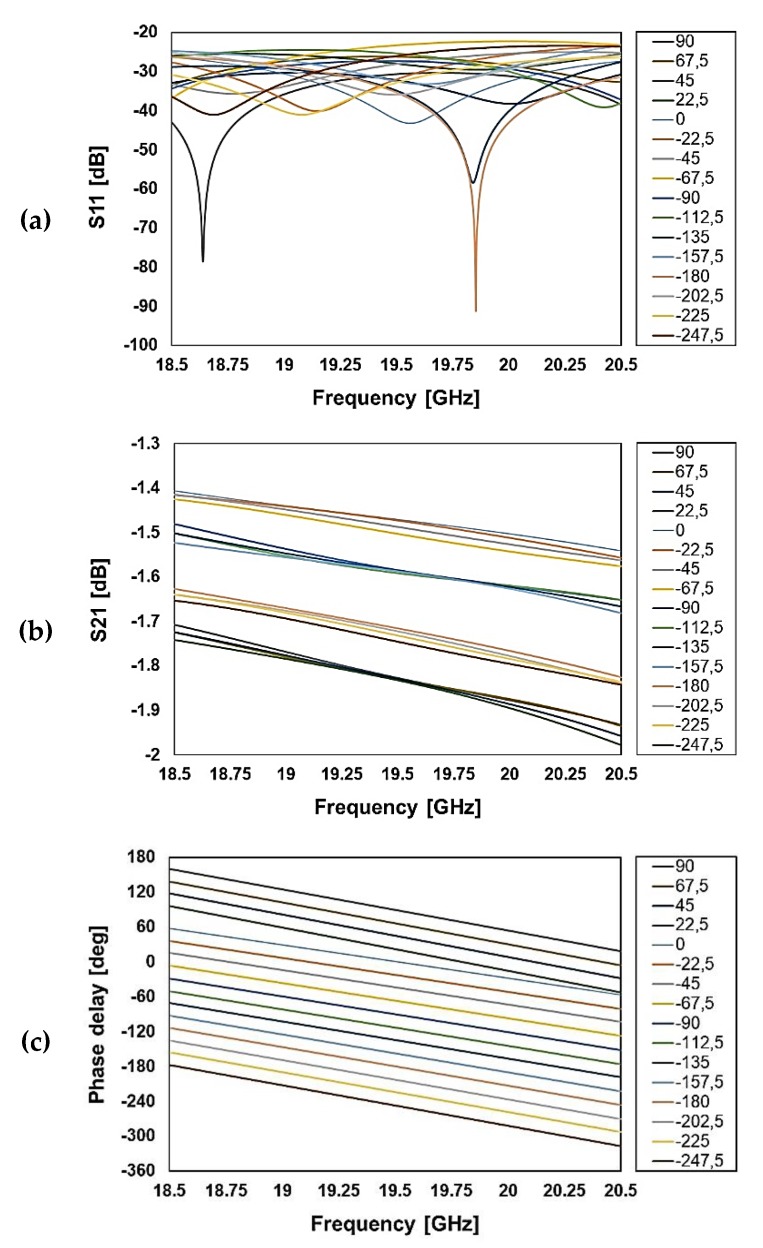
Simulated reflection (**a**), transmission (**b**), and phase shift (**c**) for several configurations of the K band RF-MEMS reconfigurable phase shifter (center frequency at 19.5 GHz).

**Figure 5 sensors-20-02612-f005:**
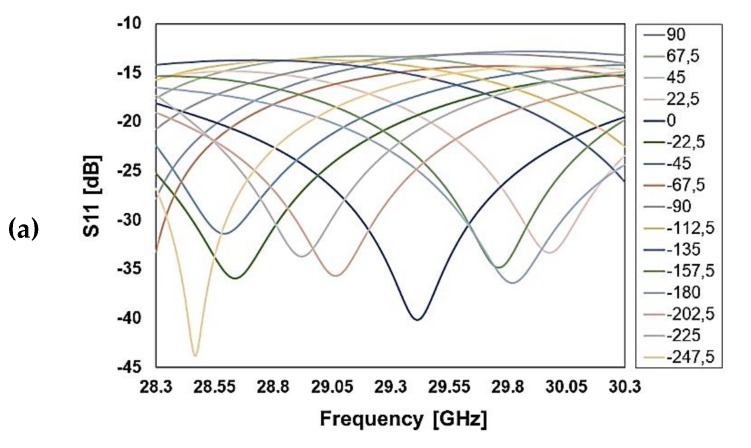
Simulated reflection (**a**), transmission (**b**), and phase shift (**c**) for several configurations of the Ka band RF-MEMS reconfigurable phase shifter (center frequency at 29.3 GHz).

**Figure 6 sensors-20-02612-f006:**
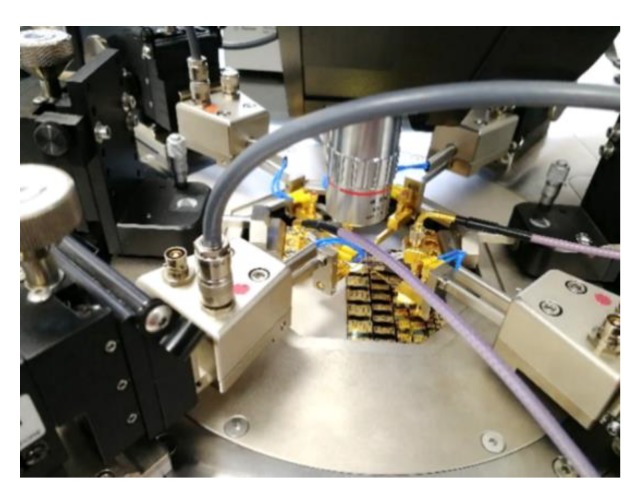
Photograph of the measurement setup featuring both RF and DC microprobes.

**Figure 7 sensors-20-02612-f007:**
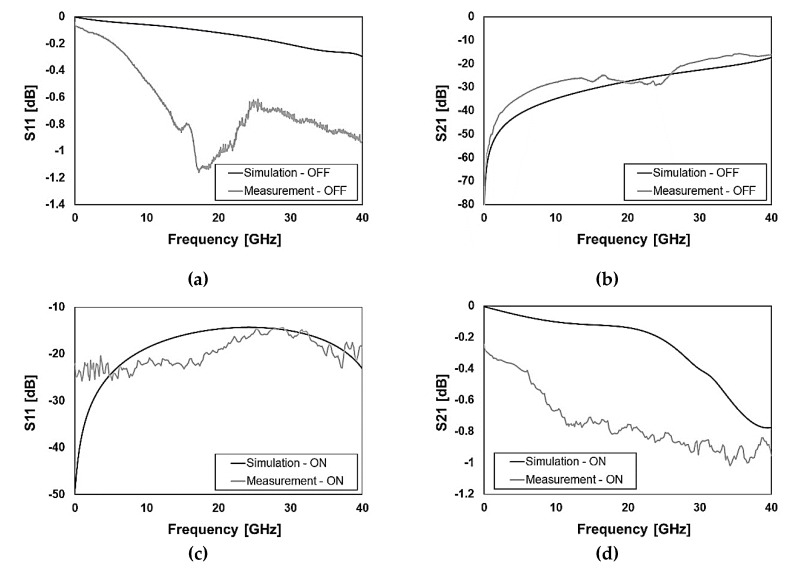
Measured versus simulated S-parameters characteristics of the transverse ohmic switch reported in Figure 2a and Figure 3a with reference to (**a**) reflection (S11) in the OFF state; (**b**) isolation (S21) in the OFF state; (**c**) reflection (S11) in the ON state; (**d**) transmission (S21) in the ON state.

**Figure 8 sensors-20-02612-f008:**
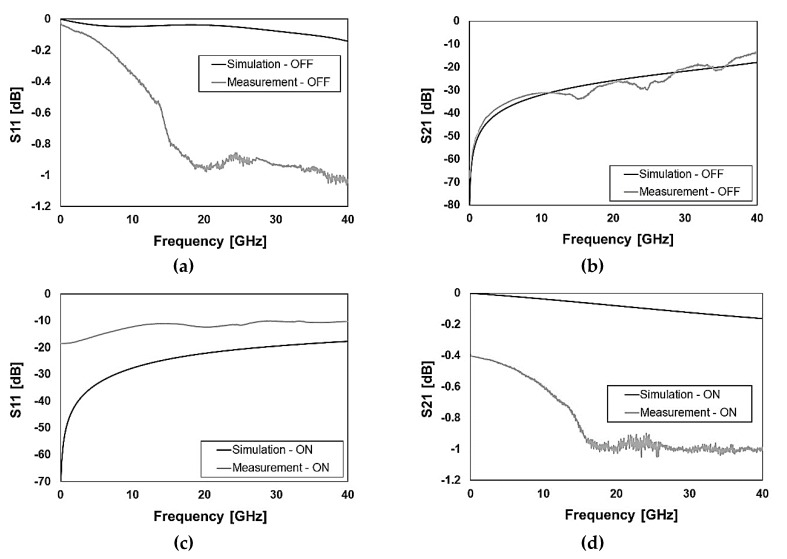
Measured versus simulated S-parameters characteristics of the longitudinal ohmic switch reported in Figure 2b and Figure 3b with reference to (**a**) reflection (S11) in the OFF state; (**b**) isolation (S21) in the OFF state; (**c**) reflection (S11) in the ON state; (**d**) transmission (S21) in the ON state.

**Figure 9 sensors-20-02612-f009:**
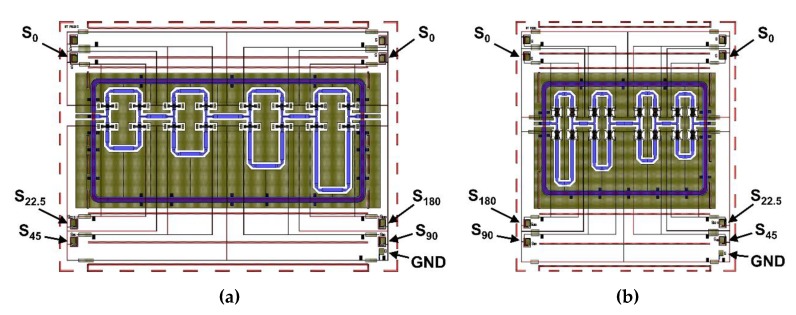
Top-view layout topology of the (**a**) K band (19.5 GHz center frequency) and of the (**b**) Ka band (29.3 GHz center frequency) RF-MEMS phase shifters design concepts.

**Figure 10 sensors-20-02612-f010:**
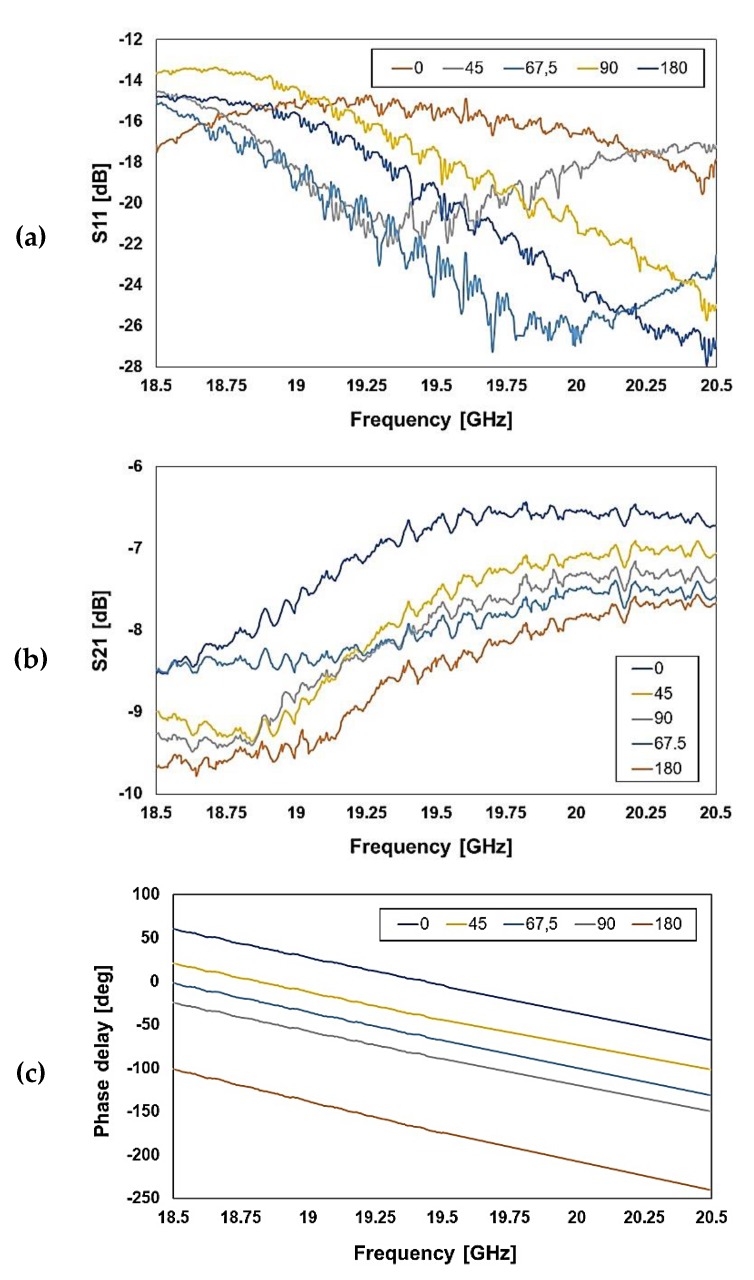
Measured characteristics of the reflection (**a**), transmission (**b**), and phase delay (**c**) for the 19.5 GHz center frequency RF-MEMS phase shifter (K band).

**Figure 11 sensors-20-02612-f011:**
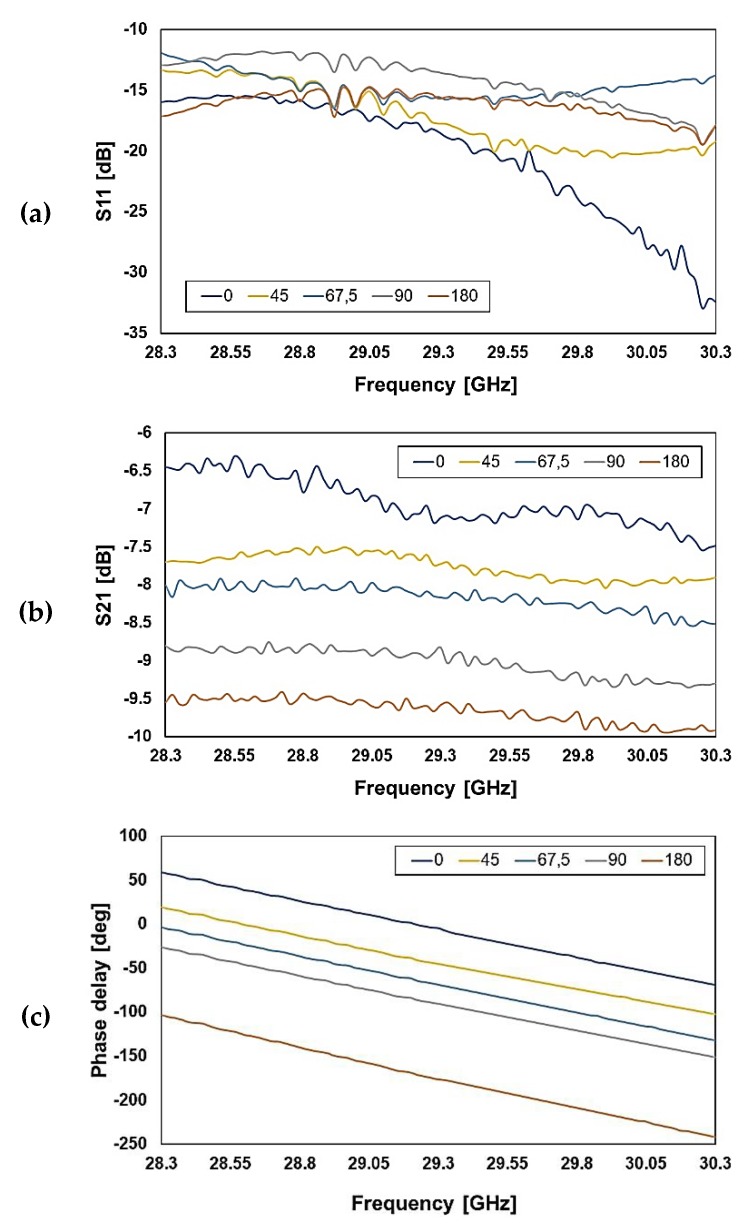
Measured characteristics of the reflection (**a**), transmission (**b**), and phase delay (**c**) for the 29.3 GHz center frequency RF-MEMS phase shifter (Ka band).

**Figure 12 sensors-20-02612-f012:**
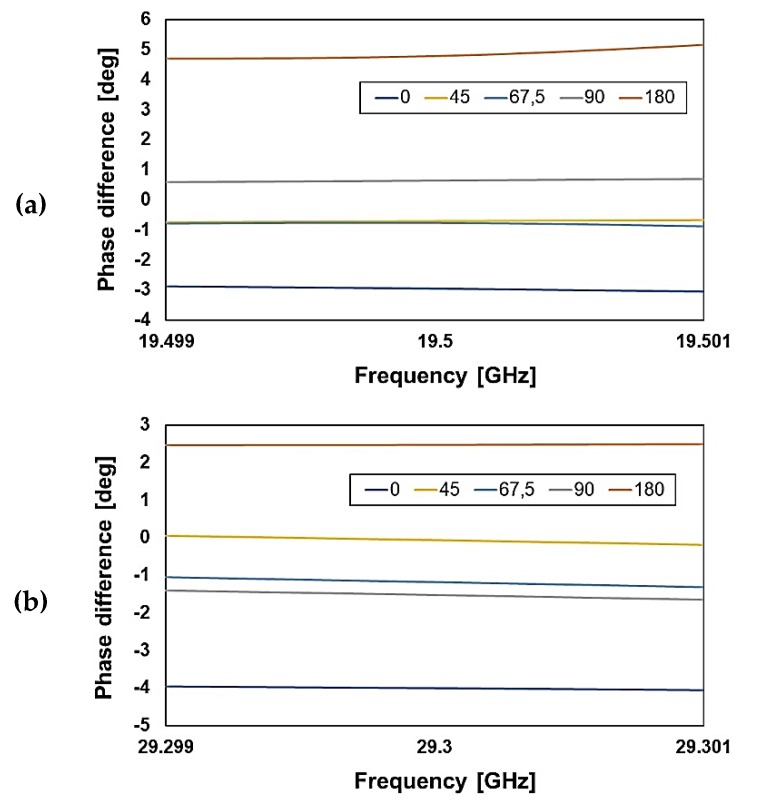
The phase difference between simulations and measurements at the center frequencies for the (**a**) K band and (**b**) Ka band RF-MEMS phase shifters.

**Table 1 sensors-20-02612-t001:** Main simulated radio frequency (RF) characteristics for the transverse and longitudinal Single Pole Single Throw (SPST) RF-MEMS switches previously shown in Figure 2.

Parameter	Transverse SPST	Longitudinal SPST
Frequency range, GHz	0–40	0–40
Actuation voltage, V	28	38
Isolation, dB	−30 @ 15 GHz −20 @ 35 GHz	−30 @ 10 GHz −20 @ 35 GHz
Insertion loss, dB	−0.8 @ 40 GHz	−0.12 @ 40 GHz

**Table 2 sensors-20-02612-t002:** Actuation (pull-in) voltage at different air-gaps for the transverse and longitudinal RF-MEMS membranes previously shown in Figure 2.

Air-Gap [µm]	*V_PI_* [V]–Transverse	*V_PI_* [V]–Longitudinal
2	28	38
2.5	37	52
2.7	42	60

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
