# Peer review of "RF-MEMS Monolithic K and Ka Band Multi-State Phase Shifters as Building Blocks for 5G and Internet of Things (IoT) Applications"

_sensors, 2020, doi:10.3390/s20092612_

Round 1

Reviewer 1 Report

The paper describes the design concept, fabrication, simulated and measured results of a multi-state phase shifter designed and manufactured in RF-MEMS technology; the phase shifters have 4 switchable stages, thus enabling 16 different phase shift configurations (4 bits), and are developed both for the K band (19.5 GHz center frequency) and Ka band (29.3 GHz center frequency).

The authors claim that the paper is focused around novel design concepts of general-purpose multi-state phase shifter but, in fact, the solution is based on a well known structure - published in many papers starting from 2001, some of them being also mentioned in the present paper ([3],[21], [22]) and part of them being used even for the K-band - and the technology used for manufacturing is a well established one (already described in [11] and [26], as well as in "RF MEMS technology for next-generation wireless communications" (Chapter 8 in "Handbook of MEMS for Wireless and Mobile Applications", ed. D. Uttamchandani, Woodhead Publishing, 2013) and in "RF-MEMS for 5G applications: a reconfigurable 8-bit power attenuator working up to 110 GHz. Part 1: design concept, technology and working principles", Microsyst. Technol. vol. 26, 675-687 (2020), published in Aug. 2019, both papers having the first author of the present paper among authors. Moreover, Fig. 6 in the present paper is the same as Fig. 1 in the paper published in Microsystem Technologies and the description is similar.

The original part of the paper is concentrated in chapter 2 (as concerns the two different types of switches each used for a specific band, and the simulated results, where simulation results for S11and S22 could be presented ans where extended attention could be given to the coupled electromechanical simulated results) and in chapter 4, where experimental results for S11and S22 should also be given (return loss diagrams), which are a measure of the design accuracy and of the accuracy of experimental realization); diagrams with phase shift errors for the two phase shifters would increase the reader's interest.

Accordingly, the paper should be restructured in order to focus on the original aspects of design, modeling and experimental results. In this respect, the Introduction chapter and the references should be drastically reduced. The authors are referring to a too wide spectrum of areas related to MEMS switches and to the reconfigurable RF structures. As a result, too many references are cited in the paper; there are many references that are not relevant for this paper [14], [17], [18], and [30] (SOLT procedure does not need a specific reference, maybe the acronym could be explained); too many review papers are present among references ([9], [12], [16], [17], [18] etc.. References to market oriented papers [4-8], [19] should be avoided, especially as these issues are well covered in other references [9-12]. Maybe the Abstract should be updated and, mandatory, the Conclusions should be also shortened and updated accordingly.

Author Response

In light of the quite pronounced criticism of Reviewer 1 in the respect of our work, we radically modified the paper. You can find below the main points of our intervention, based on your remarks, divided into modifications that concern technical aspects and the structure/cut of the paper.

************** TECHNICAL ASPECTS **************

==>> S11 (reflection) diagrams were added concerning all the simulated and measured data reported for both the RF-MEMS phase shifters design variations, along with comments and considerations on their behavior.

==>> The complete S-parameters characterization of both the standalone RF-MEMS switches was added, commenting the difference of S21 between simulations and measurements, therefore supporting the discussion on the bad results in terms of excessive attenuation of phase shifters.

==>> Phase difference diagrams were added, showing more clearly the difference between simulations and measurements.

************** STRUCTURE OF THE PAPER **************

==>> The section on the description of technology was completely removed (included old Figure 6), leaving more room for discussion of results.

==>> We addressed more effectively the novelty and differentiation of this work in comparison to existing literature, with a twofold approach. On one side we introduced explicitly the specific 5G application driving the development of the design concepts in the K and Ka band we discussed. On the other hand, we explained the novelty at design level that is mainly concerned to the RF-MEMS switching design principle adopted in the Ka band phase shifter variant.

==>> The references considered by the Reviewer 1 as redundant were removed.

==>> The references on market reports judged as inappropriate by the Reviewer 1 were removed.

==>> A few references that help better frame the existing literature were added.

==>> The abstract was modified.

==>> The introduction was significantly reduced in length and more focused on the novelty introduced by our work.

==>> Conclusions were completely rewritten.

FINAL NOTES

  1. a) All the modified parts in the paper are marked in red.

b) In order to provide a quick visual understanding of the extent of the applied modifications, a comparative PDF file was uploaded. In such a file, the original vs. current versions of the paper are compared.

Reviewer 2 Report

This manuscript reports two MEMS phase shifters of K and Ka band. The principle of TTD technology has been used in such phase shifter structures. The manuscript has done some works on the design, fabrication and tests of the MEMS switches and phase shifters. But the principle and structures of phase shifters are lack of innovations. Moreover, the article format seems to be like a draft of research report and English language is too poor.

1.Please give the summary about previous published research works of MEMS phase shifter. 2.Please give the novelty of this work compared with other published MEMS phase shifter structures. 3.Please give the simulation and test results of S11 of phase shifters, and pull-in voltages and S parameters of RF MEMS switches. 4.Please give the performance comparison of this works and previous published research works. 5.English language needs to be polished.

Author Response

We would like to thank the Reviewer 2 for carrying out the revision of this work. We radically modified the paper, accounting for the remarks emerged during the revision phase. Below you can find a point-by-point answer to all the critical aspects mentioned in your review letter.

  1. “Please give the summary about previous published research works of MEMS phase shifter.”

OUR ACTION. We extended the cited references on previously published works focused on RF-MEMS phase shifters. Moreover, we better framed the novelty and application context of our solutions, making clearer the difference with respect to already existing solutions, as better explained below.

  1. “Please give the novelty of this work compared with other published MEMS phase shifter structures.”

OUR ACTION. We addressed more effectively the novelty and differentiation of this work in comparison to existing literature, with a twofold approach. On one side we introduced explicitly the specific 5G application driving the development of the design concepts in the K and Ka band we discussed. On the other hand, we explained the novelty at design level that is mainly concerned to the RF-MEMS switching design principle adopted in the Ka band phase shifter variant.

  1. “Please give the simulation and test results of S11 of phase shifters, and pull-in voltages and S parameters of RF MEMS switches.”

OUR ACTION. The reflection (S11) parameters of the measured and simulated phase shifters were added. Moreover, we added the whole S-parameters analysis (measurements and simulations) of the standalone RF-MEMS switches, developing a proper discussion on the increased losses observed.

  1. “Please give the performance comparison of this works and previous published research works.”

OUR ACTION. This point is partially addressed by what we modified in light of the previous point 2. Moreover, we developed an extensive discussion around the causes of the bad performance in terms of losses (S21), also providing the solution to such a problem.

  1. “English language needs to be polished.”

OUR ACTION. The English written form was carefully rechecked, polishing if from a few typos and making some sentences more readable.

FINAL NOTES

  1. a) All the modified parts in the paper are marked in red.

b) In order to provide a quick visual understanding of the extent of the applied modifications, a comparative PDF file was uploaded. In such a file, the original vs. current versions of the paper are compared.

Round 2

Reviewer 1 Report

At the beginning of Chapter 2, there are mentioned too many references [9-15]. Only part of them should be kept in the paper, namely the most relevant for the present work (TTD concept, frequency ranges, etc.); most of the added references are not: [12] is referring to a much lower frequency (2.4-2.5 GHz) DTML network (not a TTD one), [13] is referring to a lower frequency (up to 10 GHz) TTD network, and [14] is referring to a lower frequency (10 GHz) optical TTD feeder, with no info regarding the MEMS swiches used.

II suggest to the authors to specify in Fig. 7 caption that it is about the switch reported in Fig. 2a and Fig. 3a (not in Fig. 2 and Fig. 3) and in Fig. 8 caption that it is about the switch reported Fig. 2b and Fig. 3b (not in Fig. 2 and Fig. 3).

In my opinion, the paper could be published now, but I believe it would have been more beneficial for the authors if it had been published after obtaining the better results they expect from the batch in progress (the values of insertion losses reported now are about 3-4 dB worse than those presented by other authors for similar structures for both frequency bands [11]).

Author Response

We thank Reviewer 1 for carrying out this further review round. Below we report the comments of the Reviewer 1 with the description, point by point, of our action in response.

The modifications in the paper are marked with VIOLET color, in order to distinguish them from the RED used in the previous round of revision.

At the beginning of Chapter 2, there are mentioned too many references [9-15]. Only part of them should be kept in the paper, namely the most relevant for the present work (TTD concept, frequency ranges, etc.); most of the added references are not: [12] is referring to a much lower frequency (2.4-2.5 GHz) DTML network (not a TTD one), [13] is referring to a lower frequency (up to 10 GHz) TTD network, and [14] is referring to a lower frequency (10 GHz) optical TTD feeder, with no info regarding the MEMS switches used.

==>> OUR ACTION. We replaced reference 12-15 with more pertinent pieces of literature, both concerning the working principle of the discussed phase shifters (i.e. TTD) and of their operating frequency ranges.

II suggest to the authors to specify in Fig. 7 caption that it is about the switch reported in Fig. 2a and Fig. 3a (not in Fig. 2 and Fig. 3) and in Fig. 8 caption that it is about the switch reported Fig. 2b and Fig. 3b (not in Fig. 2 and Fig. 3).

==>> OUR ACTION. We implemented this useful modification that allows browsing more intuitively back and forth across figures and plots.

In my opinion, the paper could be published now, but I believe it would have been more beneficial for the authors if it had been published after obtaining the better results they expect from the batch in progress (the values of insertion losses reported now are about 3-4 dB worse than those presented by other authors for similar structures for both frequency bands [11]).

==>> OUR ACTION. We understand the rationale behind this comment and from the conceptual point of view we agree on it. However, given the prolonged and still ongoing lockdown emergency condition due to the Covid-19 breakout (both in Italy and in the Russian Federation), we are unable to say when we will be able to access again our facility to complete the processing. After that, we cannot predict when we might perform additional S-parameters characterization, as we have to access a third-party facility. In light of this situation, we would prefer to publish a.s.a.p. the current results, despite they are not as good as we expected, and reserve to a future publication the presentation of improved experimental characteristics.

Reviewer 2 Report

Authors did not give the proper response of my questions, especially the novelty of this work and the performance comparison of this work and other previous works.

Author Response

We would like to thanks Reviewer 2 for this additional round of revision. We added in the introduction an additional paragraph that we hope is more effective in farming the motivation behind our work and where the aspects of novelty are nested.

The modifications in the paper are marked with VIOLET color, in order to distinguish them from the RED used in the previous round of revision.

Round 3

Reviewer 2 Report

I think the quality of this manuscript is enough to be published in SENSORS.